# Revisiting Cooperative Off-Policy Multi-Agent Reinforcement Learning

Yueheng Li [1]  Guangming Xie [1 2]  Zongqing Lu [3]

## Abstract

Cooperative Multi-Agent Reinforcement Learning (MARL) has become a critical tool for addressing complex real-world problems. However, off-policy MARL methods, which rely on joint Q-functions, face significant scalability challenges due to the exponentially growing joint action space. In this work, we highlight a critical yet often overlooked issue: erroneous Q-target estimation, primarily caused by extrapolation error. Our analysis reveals that this error becomes increasingly severe as the number of agents grows, leading to unique challenges in MARL due to its expansive joint action space and the decentralized execution paradigm. To address these challenges, we propose a suite of techniques tailored for off-policy MARL, including annealed multi-step bootstrapping, averaged Q-targets, and restricted action representation. Experimental results demonstrate that these methods effectively mitigate erroneous estimations, yielding substantial performance improvements in challenging benchmarks such as SMAC, SMACv2, and Google Research Football.

## 1. Introduction

Cooperative Multi-Agent Reinforcement Learning (MARL) has proven to be a powerful framework for tackling a diverse range of complex real-world challenges, including autonomous driving (Zhou et al., 2020), traffic management (Singh et al., 2020), and robot swarm coordination (Hüttenrauch et al., 2019; Zhang et al., 2021a). However, the inherent complexity of these scenarios introduces significant challenges, particularly with scalability, as the joint action space grows exponentially with the number of agents.

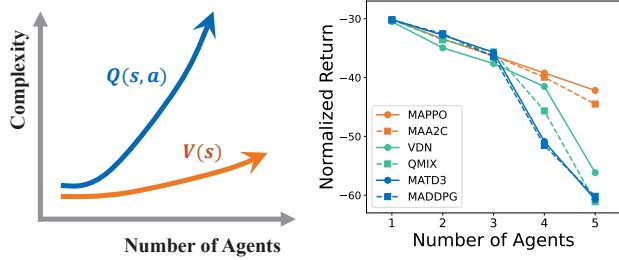

Figure 1. Comparison between off-policy and on-policy MARL methods

Off-policy MARL methods, which require the estimation of joint Q-functions, are especially affected by this exponential growth. As illustrated in Fig. 1 (left), the computational complexity of off-policy methods increases substantially with the number of agents, in contrast to on-policy methods, which estimate the joint state value function directly. Although recent approaches, such as value factorization methods (Son et al., 2019; Sunehag et al., 2018; Rashid et al., 2020b; Wang et al., 2021b), have shown promise in addressing this scalability issue, their performance often deteriorates as the number of agents grows. This limitation is evident in Fig. 1 (right), which compares popular off-policy and on-policy MARL methods on the MPE (Lowe et al., 2017) *spread* task. While both categories of methods achieve comparable performance for smaller agent populations ($N \leq 3$), off-policy methods experience a significant performance drop as the number of agents increases. Even value factorization techniques, though somewhat more scalable, fail to address this degradation fully.

In this paper, we investigate the root causes behind the limitations of off-policy MARL methods. Our analysis identifies a critical yet often overlooked issue: estimation errors in the Temporal Difference (TD) target. These errors stem from two MARL-specific challenges: (1) the exponentially expanding joint action space amplifies extrapolation errors in TD target estimation, and (2) addressing these errors necessitates strict monotonicity in value factorization methods.

Building on these observations, we propose a suite of techniques to mitigate estimation errors in off-policy MARL, including annealed multi-step bootstrapping, averaged TD targets, and restricted action representations. These techniques improve performance by leveraging temporally ex-

[1]College of Engineering, [2]Institute of Artificial Intelligence, [3]School of Computer Science, Peking University. Correspondence to: Guangming Xie <xiegming@pku.edu.cn>, Zongqing Lu <zongqing.lu@pku.edu.cn>.

*Proceedings of the $42^{nd}$ International Conference on Machine Learning*, Vancouver, Canada. PMLR 267, 2025. Copyright 2025 by the author(s).

tended trajectories to reduce bias, averaging target values to lower variance, and simplifying target estimation to alleviate extrapolation errors. When integrated into existing off-policy MARL methods, these techniques yield substantial performance gains across a variety of challenging tasks, including SMAC (Samvelyan et al., 2019), SMACv2 (Ellis et al., 2023), and Google Research Football (GRF) (Kurach et al., 2020).

## 2. Background

### 2.1. Dec-POMDP and CTDE

We consider Decentralized Partially Observable Markov Decision Process (Dec-POMDP) (Oliehoek & Amato, 2016) in modeling cooperative multi-agent tasks. The Dec-POMDP is characterized by the tuple $\langle \mathcal{N}, \mathcal{S}, \mathcal{A}, r, \mathcal{P}, \mathcal{O}, \mathcal{Z}, \gamma \rangle$, where $\mathcal{N}$ is the set of agents, $\mathcal{S}$ is the set of states, $\mathcal{A}$ is the set of actions, $r$ is the reward function, $\mathcal{P}$ is the transition probability function, $\mathcal{Z}$ is the individual partial observation generated by the observation function $\mathcal{O}$, and $\gamma$ is the discount factor. At each timestep, each agent $i \in \mathcal{N}$ receives a partial observation $o_i \in \mathcal{Z}$ according to $\mathcal{O}(s; i)$ at state $s \in \mathcal{S}$. Then, each agent selects an action $a_i \in \mathcal{A}$ according to its action-observation history $\tau_i \in (\mathcal{Z} \times \mathcal{A})^*$, collectively forming a joint action denoted as $\boldsymbol{a}$. The state $s$ undergoes a transition to the next state $s'$ in accordance with $\mathcal{P}(s'|s, \boldsymbol{a})$, and agents receive a shared reward $r$. The joint action-value function is expressed as $Q^\pi(s_t, a_t) = \mathbb{E}_{s_{t+1:\infty}, a_{t+1:\infty}} \left[ \sum_{i=0}^\infty \gamma^i r_{t+i} \right]$, where $\pi$ denotes the joint policy.

This work adheres to the Centralized Training with Decentralized Execution (CTDE) (Oliehoek et al., 2008; Kraemer & Banerjee, 2016) paradigm. In the training phase, CTDE enables policy training to capitalize on globally available information and facilitates the exchange of information among agents. Conversely, during the execution phase, each agent is restricted to accessing solely its individual action-observation history, thereby embodying decentralized execution principles.

### 2.2. Value-Based RL

Value-based RL methods typically involve the iterative adjustment of Q-functions based on the Bellman equation: $Q_{k+1} = \mathcal{T}^\pi Q_k = r + \gamma \mathcal{P}^\pi Q_k$, where $\mathcal{T}^\pi$ denotes the Bellman operator and $\mathcal{P}^\pi Q = \sum_{s'} \mathcal{P}(s'|s, \boldsymbol{a}) \sum_{\boldsymbol{a}'} \pi(\boldsymbol{a}'|s) Q(s, \boldsymbol{a}')$. Restricting the policy to be greedy w.r.t the current Q-function, i.e., $\pi \in \boldsymbol{G}(Q)$, where $\boldsymbol{G}(Q)$ is the set of all greedy policies w.r.t $Q$, transforms the operator into the Bellman optimality operator $\mathcal{T}$, resulting in the Q-learning update $Q_{k+1} = \mathcal{T} Q_k$.

In scenarios with a large state space, the value is often estimated using a differentiable function approximator

$Q(s, \boldsymbol{a}; \theta)$ parameterized by $\theta$. Within the framework of deep Q-learning, updates depend on a batch of transitions $(s, \boldsymbol{a}, r, s')$ derived from the replay buffer $\mathcal{D}$. The training of the value function aims to minimize the mean square error:

$$L(\theta) = \mathbb{E}_{(s, \boldsymbol{a}, r, s') \sim \mathcal{D}} \left[ (Q(s, \boldsymbol{a}; \theta) - y)^2 \right], \quad (1)$$

where $y = r + \gamma \max_{\boldsymbol{a}'} Q(s', \boldsymbol{a}'; \theta')$ represents the TD target. The function $Q(\cdot; \theta')$ corresponds to the target network parameterized by $\theta'$. The periodically updated $\theta'$ ensures a consistent target across multiple iterations.

### 2.3. Value Factorization

Value factorization methods involve learning a factorized value function that encompasses per-agent utilities, denoted as $[Q_i(\tau_i, a_i)]_{i=1}^n$, and is rooted in the principles of Q-learning. A prominently discussed concept in this context is Individual-Global-Max (IGM) (Son et al., 2019), designed to ensure that the locally greedily selected action aligns with the jointly optimal action. Adhering to this constraint, various value factorization methods have been proposed, with some notable examples being VDN (Sunehag et al., 2018), QMIX (Rashid et al., 2020b), QTRAN (Son et al., 2019), and QPLEX (Wang et al., 2021b). In the representation of the joint Q-function, VDN employs an additive assumption: $Q(s, \boldsymbol{a}) = \sum_{i=1}^n Q_i(\tau_i, a_i)$. On the other hand, QMIX utilizes a monotonic mixing function

$$Q(s, \boldsymbol{a}) = f(s, Q_1(\tau_1, a_1), ..., Q_n(\tau_n, a_n)) \text{ with } \frac{\partial f}{\partial Q_i} \geq 0, \tag{2}$$

where the function $f$ is approximated using a hypernetwork that takes the global state $s$ as input and produces nonnegative weights, ensuring monotonicity. QPLEX extends the factorization to enable greater representational capacity via a duplex advantage-weighted formulation:

$$Q(s, \boldsymbol{a}) = \sum_i (\lambda_i(\tau, \boldsymbol{a}) - 1) A_i(\tau, a_i) + \sum_i Q_i(\tau, a_i), \quad (3)$$

where $A_i(\tau, a_i) = Q_i(\tau, a_i) - \max_{a_i'} Q_i(\tau, a_i')$.. The key to QPLEX's full expressiveness lies in the weight parameter $\lambda_i(\tau, \boldsymbol{a}) > 0$, which is conditioned on the joint action space.

For consistency throughout the remainder of this paper, we denote individual Q-functions as $Q_i(s, a_i)$, even though some methods use $\tau_i$ or $\tau$ as inputs.

## 3. Analysis on Off-Policy MARL

In this section, we investigate the challenges that hinder the performance of off-policy MARL algorithms. We begin with an error decomposition of the Q-function, followed by identifying two MARL-specific issues related to the estimation errors of the TD target.

### 3.1. Error Decomposition of Q-function

In practical deep Q-learning with function approximation, the learning of Q-functions may encounter various errors. Following the definition from Anschel et al. (2017), the error $\Delta$ between the current value and the optimal value can be decomposed into three terms:

$$
\begin{aligned}
\Delta &= Q(s, \boldsymbol{a}; \theta) - Q^*(s, \boldsymbol{a}) \\
&= \underbrace{Q(s, \boldsymbol{a}; \theta) - y_{s,\boldsymbol{a}}}_{\text{TAE}} + \underbrace{y_{s,\boldsymbol{a}} - \hat{y}_{s,\boldsymbol{a}}}_{\text{TEE}} + \underbrace{\hat{y}_{s,\boldsymbol{a}} - Q^*(s, \boldsymbol{a})}_{\text{OD}},
\end{aligned}
\tag{4}
$$

where $y_{s,\boldsymbol{a}} = \mathbb{E}_{\mathcal{D}}\left[r + \gamma \max_{\boldsymbol{a}'} Q(s', \boldsymbol{a}'; \theta)\right]$ is the estimated target and $\hat{y}_{s,\boldsymbol{a}} = \mathbb{E}_{\mathcal{D}}\left[r + \gamma \max_{\boldsymbol{a}'} Q(s', \boldsymbol{a}'; \hat{\theta})\right]$ represents the true target with $\hat{\theta} = \arg\min_{\theta} \mathbb{E}_{\pi}[(Q(s, \boldsymbol{a}; \theta) - y_{s,\boldsymbol{a}})^2]$.

**Target Approximation Error (TAE)** captures the discrepancy between the learned $Q(s, \boldsymbol{a}; \theta)$ and its target $y_{s,\boldsymbol{a}}$. This error can be attributed to factors such as the inexact minimization of $\theta$ through gradient descent and the limited representational capacity of neural networks. **Target Estimation Error (TEE)** measures the difference between the estimated target $y_{s,\boldsymbol{a}}$ and the true target $\hat{y}_{s,\boldsymbol{a}}$, which can be influenced by issues such as overestimation and extrapolation errors. It is important to note that prior work often refers to this term as "overestimation error". However, overestimation assumes that the error on the Q-target follows a uniform distribution. In this paper, we use the term TEE to more accurately reflect that the error originates from the Q-target itself, which is typically not uniformly distributed and may be dominated by extrapolation errors in MARL (see Sec. 3.2). **Optimality Difference (OD)** quantifies the gap between the value function of the current policy and that of the optimal policy. Unlike TAE and TEE, which depend on the current Q-function approximation with $\theta$, OD pertains solely to how well the value function fits the Bellman optimality equation.

Previous research in MARL has often focused on TAE, exploring ways to improve target fitting. For example, in transitioning from a centralized Q-function to a factorized Q-function, the function $Q(s, \boldsymbol{a}; \theta)$ is simplified as a function of $Q_i(s, a_i; \theta_i)$, which facilitates target approximation and reduces TAE. However, early value factorization methods were found to be insufficiently expressive, meaning that they could not minimize TAE to zero. As a result, much of the research on value factorization has concentrated on improving the representational capacity of the Q-function. Nevertheless, recent empirical studies (Yu et al., 2022; Ellis et al., 2023; Hu et al., 2023) suggest that improving representational capacity alone does not always lead to performance gains, particularly in complex domains.

The core issue lies in the tendency of prior research to simplify Q-function learning as a regression problem or to focus solely on theoretical properties in single-step matrix games (Mahajan et al., 2019; Son et al., 2019; Wang et al., 2021b;c; Rashid et al., 2020a). This simplification turns the problem into one of Q-function updating toward the true target, which mainly involves TAE, while overlooking the impact of TEE, the second term in Eq. (4).

In this paper, we identify that TEE is a crucial issue for the performance of off-policy MARL algorithms, primarily driven by **extrapolation error**. While both TAE and TEE capture Q-function estimation errors, TEE specifically reflects the accuracy of predictions for $Q(s', \boldsymbol{a}')$, whereas TAE pertains to $Q(s, \boldsymbol{a})$. This distinction is critical because $\boldsymbol{a}'$ may not be observed in past trajectories, which can lead to significant extrapolation errors when predicting its value.

### 3.2. Extrapolation Error in MARL

Extrapolation error arises in RL when the value function inaccurately estimates the value of actions that are unseen or rare (**?**). If a state-action pair $(s, \boldsymbol{a})$ is absent from the dataset, the Q-function, $Q(s, \boldsymbol{a}; \theta)$, produces an uncertain prediction due to the neural network's inability to generalize accurately for unseen combinations. Specifically, in the Q-function update, while the transition $(s, \boldsymbol{a}, r, s')$ is sampled from the dataset, the next action $\boldsymbol{a}'$ generated by the Q-function or policy may be unseen or rare. This can lead to significant errors in estimating $Q(s', \boldsymbol{a}')$, propagating poor estimates to subsequent values.

Extrapolation error is most commonly associated with offline RL (Fujimoto et al., 2019; Levine et al., 2020), where the agent operates on a fixed dataset and cannot interact further with the environment, increasing the likelihood that the policy will sample actions not present in the dataset. In contrast, online RL allows the policy to interact with the environment and collect new data for the actions it generates, enabling self-correction and reducing extrapolation error. However, in MARL, extrapolation error remains a significant issue even in online settings. There are two main reasons for this. First, the exponentially growing joint action space makes the network increasingly susceptible to erroneous extrapolations, even in online RL (Fujimoto et al., 2023). Second, value factorization has a different property related to extrapolation error, which we will discuss in the next subsection.

For example, consider 5 agents with 10 possible actions each. The joint action space would then contain $10^5$ possible action combinations. To cover all possible joint actions for a specific state using uniform sampling, over $10^6$ samples would be required. In practice, this is rarely achievable with limited samples, meaning the Q-target $Q(s', \boldsymbol{a}')$ must often be estimated via extrapolation. It is noteworthy that extrapolation error is more severe for actions than for states. States tend to have more similarity in Euclidean space, making it easier for neural networks to extrapolate. In contrast,

discrete actions, often represented as one-hot vectors, lack such Euclidean similarity, which makes extrapolation difficult. This issue is especially pronounced in MARL, where the joint action space grows exponentially, increasing the challenge of accurately estimating Q-values.

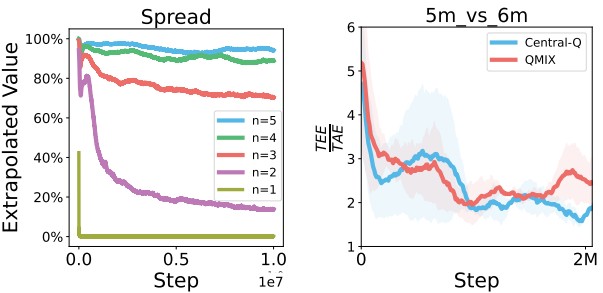

*Figure 2.* (left) Proportion of extrapolated values for tasks with different numbers of agents. (right) TEE/TAE comparison for centralized Q-function and QMIX.

To illustrate this issue, we present experimental results on the MPE and SMAC environments. Fig. 2(left) shows the proportion of extrapolated values in tasks with varying numbers of agents. The proportion is calculated based on the fraction of $(s', \boldsymbol{a}')$ pairs in each update that are absent from the replay buffer. As shown, in the single-agent case, extrapolation is not a significant issue, as the proportion of extrapolated values quickly decreases to zero. However, as the number of agents increases, the exponentially growing joint action space leads to a substantial reliance on extrapolated Q-targets. Fig. 2(right) compares TEE and TAE for a centralized Q-function and QMIX. As observed, TEE is larger than TAE and does not decrease significantly over time, as the large joint action space requires an impractically high number of samples to cover all possible actions. Moreover, although factorized Q-functions like QMIX demonstrate improved scalability over centralized Q-functions (as shown in Fig. 1), they do not resolve the TEE issue. This indicates that extrapolation error is a common challenge for all existing off-policy MARL algorithms.

### 3.3. Error Propagation of TEE

Another challenge in online MARL is that the joint Q-function does not directly dictate the behavior of individual policies in value factorization methods, distinguishing it from single-agent RL.

As discussed earlier, online RL can mitigate extrapolation errors by increasing the frequency of executing unseen actions in the environment. However, in online MARL, individual policies are derived from individual utilities rather than directly from the joint Q-function. Specifically, overestimation of the joint Q-function does not necessarily increase the probability of selecting the corresponding actions for individual agents, rendering it ineffective at addressing extrapolation error. To tackle this, we introduce the concept of Error Propagation Consistency (EPC):

**Definition 3.1** (EPC). *In value factorization methods, for a joint value function $Q(s, \boldsymbol{a})$, if its corresponding individual utilities $[Q_i(s, a_i)]_{i=1}^n$ satisfy that, overestimation of $Q(s, \boldsymbol{a})$ will result in the overestimation of all $Q_i(s, a_i)$, we say that the factorization structure adheres to EPC.*

EPC is critical for mitigating TEE in value factorization methods. More importantly, if techniques are applied to reduce the error in the joint Q-function, EPC ensures that these improvements propagate to the individual utilities. This principle forms the basis for addressing TEE in the next section.

In essence, factorization structures adhering to EPC must exhibit monotonicity, as stated in the following proposition:

**Proposition 3.2.** *Monotonicity, expressed as $\frac{\partial Q(s, \boldsymbol{a})}{\partial Q_i(s, a_i)} \geq 0$, stands as a sufficient and necessary condition for EPC.*

To illustrate, consider gradient descent on the objective function $\min \mathbb{E}[(y - f(Q_1, ..., Q_n))^2]$, where $y$ represents the TD target and $Q = f(Q_1, ..., Q_n)$ denotes the factorized joint value function. The update rule of individual utility on $(s, \boldsymbol{a})$ is given by:

$$Q_i(s, a_i) \leftarrow Q_i(s, a_i) +$$
$$2\alpha[y - f(s, Q_1(s, a_1), ..., Q_n(s, a_n))]\frac{\partial f}{\partial Q_i}|_{s, \boldsymbol{a}},$$

where $\alpha$ is the learning rate. For a monotonic $f$ with $\frac{\partial f}{\partial Q_i} \geq 0$, if the target $y$ exceeds the current value function, i.e., $y > f(Q_1, ..., Q_n)$, the individual utility $Q_i$ will increase for $(s, a_i)$. This means that overestimation of the target Q-function leads to overestimation of all individual utilities $Q_i(s, a_i)$. Since actions are selected individually based on $Q_i(s, a_i)$, this overestimation increases the probability of selecting $a_i$ for each $i$, thereby making the joint action $\boldsymbol{a} = [a_1, ..., a_n]$ under state $s$ more likely. This results in the corresponding trajectory with poor estimates being revisited more frequently, correcting the erroneous estimation.

Conversely, for a non-monotonic function, a larger target $y$ may lead to smaller values for some $Q_i$. Consequently, the likelihood of selecting the joint action $\boldsymbol{a}$ may decrease because underestimation of $Q_i(s, a_i)$ lowers the probability of selecting $a_i$. In such cases, the erroneous estimation cannot be corrected, and consistently using this erroneous value as the target exacerbates error accumulation. Similar observations are reported in Peng et al. (2021) and Hu et al. (2023), where non-monotonic factorization, despite its better expressiveness, tends to underperform compared to monotonic counterparts, particularly in complex tasks.

# 4. Addressing TEE

In this section, we introduce several techniques to mitigate TEE. As discussed previously, TEE arises primarily from extrapolation over the large joint action space in $Q(s', \boldsymbol{a}')$. EPC provides a mechanism to directly control the joint Q-function error in value factorization methods. Consequently, we propose the following approaches to address TEE: 1) Reduce reliance on approximated $Q(s', \boldsymbol{a}')$ by employing its Monte Carlo estimation instead. 2) Mitigate uncertainty in rarely observed inputs by learning multiple $Q(s', \boldsymbol{a}')$ functions and averaging them. 3) Restrict the representation of joint actions to reduce their impact. These techniques are generally applicable across existing off-policy MARL methods. In the following subsections, we elaborate on the methods inspired by these strategies.

## 4.1. Annealed Multi-Step Bootstrapping

To reduce reliance on $Q(s', \boldsymbol{a}')$ in TD targets, we propose replacing it with multi-step returns (Sutton & Barto, 2018). Multi-step targets derived from unbiased samples can offset the effects of an undertrained value network, thereby reducing estimation error.

A prominent multi-step method for Q-learning is Peng's $Q(\lambda)$ (PQL) (Peng & Williams, 1994). The PQL operator, $\mathcal{N}_\lambda^{\mu,\pi}$ applicable to any policies $\mu$ and $\pi$, is defined as:

$$\mathcal{N}_\lambda^{\mu,\pi} Q = (1-\lambda) \sum_{n=1}^\infty (\lambda \mathcal{T}^\mu)^{(n-1)} \mathcal{T}^\pi Q, \qquad (5)$$

where $\lambda \in [0,1]$. PQL is widely adopted in existing MARL algorithms (Peng et al., 2021; Zhang et al., 2021b; Hu et al., 2023), yet its properties remain underexplored and merit further examination.

Considering the error at iteration $k$ is $e_k$, we consider the following update with error propagation:

$$\pi_k \in \boldsymbol{G}(Q_k) \text{ and } Q_{k+1} := \mathcal{N}_\lambda^{\mu,\pi_k}(Q_k + e_k), \qquad (6)$$

where $\mu$ is maintained as a fixed behavior policy (discussed later). The following proposition illustrates PQL's error-reducing properties:

**Proposition 4.1.** *The target estimation error for each update step $k$ satisfies:*

$$\|Q_{k+1} - \mathcal{N}_\lambda^{\mu,\pi_k} Q_k\|_\infty \leq \beta \epsilon, \qquad (7)$$

*where $\epsilon = \|e_k\|_\infty$ and $\beta = \frac{\gamma(1-\lambda)}{1-\gamma\lambda}$. The accumulated error related to $\epsilon$ is $\mathcal{O}(\frac{\gamma(1-\lambda)}{(1-\gamma)^2}\epsilon)$.*

*Proof.* The proof follows from Kozuno et al. (2021); full details are provided in Appendix B. □

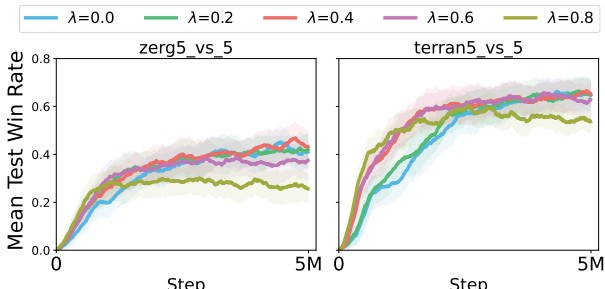

*Figure 3.* QMIX'S performance on SMACv2 with different $\lambda$.

This proposition establishes that TEE and its propagation depend on $\lambda$. A larger $\lambda$ reduces error, consistent with the intuition that $\lambda$ balances collected returns and learned value functions in target estimation.

However, a larger $\lambda$ is not always beneficial. With a fixed behavior policy $\mu$, the PQL operator converges to a biased policy (Harutyunyan et al., 2016). While convergence to the optimal policy is possible as $\mu$ approaches $\pi$ (Kozuno et al., 2021), this requires extensive timesteps and may not hold in practice. As shown in Fig. 3, while larger $\lambda$ values improve initial training efficiency, their performance may degrade over time, ultimately lagging behind smaller $\lambda$ values.

To capitalize on PQL's initial efficiency while avoiding biased final performance, we propose a $\lambda$ annealing strategy. This approach leverages PQL's error-reducing properties at the start of training with a large initial $\lambda$, then gradually anneals $\lambda$ to prevent policy bias. Specifically, we anneal $\lambda$ from 1 to $\lambda^*$ over the course of training, where $\lambda^*$ is a hyperparameter. The detailed annealing scheme is provided in Appendix D.1.

## 4.2. Averaged TD Target

To directly reduce TEE, a common approach involves using multiple independent Q-function estimators and applying ensemble methods. Ensemble techniques are well-known for reducing variance and improving robustness (Ganaie et al., 2022).

Consider $M$ independently estimated Q-functions: $Q(s, \boldsymbol{a}; \theta^j) = Q^*(s, \boldsymbol{a}) + e^j(s, \boldsymbol{a})$, with $e^j$ representing the error term, assumed to be i.i.d across $j$ for each fixed state-action pair. By averaging these Q-functions, the variance of the target estimation is reduced proportionally to $1/M$:

$$\text{Var}[\frac{1}{M} \sum_{j=1}^M Q(s, \boldsymbol{a}; \theta^j)] = \frac{1}{M} \text{Var}[e^j(s, \boldsymbol{a})]. \qquad (8)$$

This reduction leverages the i.i.d. nature of the error $e^j$.

Unlike prior works that impose additional assumptions on

error properties (Anschel et al., 2017; Chen et al., 2021), we do not make such assumptions. Instead, we acknowledge that the limited expressiveness of value factorization methods can introduce model bias into $e^j$, stemming from TAE, as discussed in Section 3. Additionally, due to these biases, we avoid using conservative methods like those employed in offline RL (An et al., 2021; Bai et al., 2022). Further details are provided in Appendix D.2.

In this approach, we propose directly averaging Q-functions within the TD target, incorporating the action space. Specifically: for methods that only utilize joint Q-functions, we utilize $M$ joint Q-function: $Q(s, \boldsymbol{a}; \bar{\theta}) = \sum_{j=1}^{M} Q(s, \boldsymbol{a}; \theta^j)$; for value factorization methods, we utilize $M$ individual utility: $Q(s, \boldsymbol{a}; \bar{\theta}, \psi) = \sum_{j=1}^{M} Q(s, [Q_i(s, a_i; \theta_i^j)]; \psi)$, where $\psi$ represents the parameter of the mixing network. The corresponding loss function can be written as:

$$L(\boldsymbol{\theta}, \psi) = \sum_{j=1}^{M} \mathbb{E}_{\mathcal{D}} \big[ (Q(s, \boldsymbol{a}; \boldsymbol{\theta}^j, \psi) - y_{s, \boldsymbol{a}})^2 \big], \quad (9)$$

where $y_{s, \boldsymbol{a}}$ is the PQL target using averaged Q-function.

For value factorization methods like QMIX, it is worth noting that there is no need to average multiple mixing networks, as they do not depend on the action space. Further details are provided in Appendix D.2. Another critical point is that the theoretical guarantees underlying these results implicitly assume that the factorization satisfies the EPC condition, i.e., monotonicity. This requirement is essential because both PQL and ensemble methods aim to control the error of the joint Q-function, and only a monotonic factorization can propagate this control to individual utilities effectively.

### 4.3. Restricted Action Representation

Since different methods utilize the joint action space in varying ways, we illustrate the application of the Restricted Action Representation (RAR) technique using MADDPG and QPLEX as examples. While the specific methods differ, they share the common objective of reducing the influence of joint actions in the TD target.

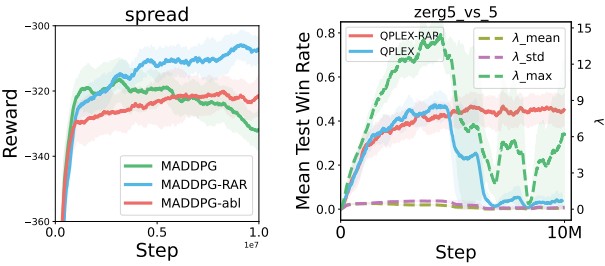

For methods that directly use the joint Q-function, such as MADDPG, we apply a function $g$ to map the joint action space to a smaller, more manageable space. The $g(\boldsymbol{a})$ is modeled as a neural network that takes the joint action $\boldsymbol{a}$ as input and outputs a vector of categorical variables. This network is optimized using straight-through gradients (Bengio et al., 2013). For example, in MPE with 4 agents, the joint action space has a size of $5^4$. By using the mapping function $g(\boldsymbol{a})$, we reduce this space to $2^4$ through a 10-dimension vector. This vector is comprised of 5 categorical distributions, each with 2 possible classes. We then sample from $g(\boldsymbol{a})$ and convert the result into a one-hot vector. The target Q-function is then given by: $Q(s', \text{onehot}(\text{draw}(g(\boldsymbol{a}'))))$. This ensures that, instead of having $5^4$ Q-values at each state $s'$, we only have $2^4$ values, thus reducing the extrapolation in the target Q-function.

The corresponding results are shown in Fig. 4(left), where the "MADDPG-abl" curve represents the ablation of using $Q(s', g(\boldsymbol{a}'))$, highlighting that simply mapping the joint action to $g(\boldsymbol{a})$ does not lead to performance improvement. The key reason is that mapping the action space to categorical $g(\boldsymbol{a})$ restricts the representation of the joint action while continuous $g(\boldsymbol{a})$ cannot.

For methods like QPLEX, which utilize joint actions as an individual component, we apply a simpler technique to restrict the influence of joint actions. QPLEX relies on a weight parameter $\lambda_i(s, \boldsymbol{a})$, which can become erroneous due to the vast size of the joint action space. As shown in Fig. 4(right), the dashed line and the right y-axis display the mean, std and maximum values of QPLEX's $\lambda_i$ during training. We observe that while the mean and std of $\lambda_i$ remain small and stable, the maximum value grows significantly over time. This indicates the presence of outliers in $\lambda_i$ likely due to error accumulation during training. This error accumulation leads to instability around the 5M training step, causing a significant drop in QPLEX's performance.

The reason for this issue is straightforward: as $\lambda_i$ grows large at rarely encountered state-action pairs $(s, \boldsymbol{a})$, erroneous values accumulate over time. This error persists because the probability of selecting action $\boldsymbol{a}$ does not increase with $\lambda_i(s, \boldsymbol{a})$. This is similar to the issue discussed in Section 3.3. To mitigate this, we propose directly restricting $\lambda_i$ in QPLEX. Specifically, we bound it by applying the Sigmoid function: $\lambda_i^*(s, \boldsymbol{a}) = \text{Sigmoid}(\lambda_i(s, \boldsymbol{a}))$, which ensures that $\lambda_i^* \in [0, 1]$. The results in Fig. 4(right) show that QPLEX-RAR significantly improves the stability of QPLEX, demonstrating that RAR effectively reduces the negative impact of the joint action space.

*Figure 4.* The effect of RAR on MADDPG (left) and QPLEX (right).

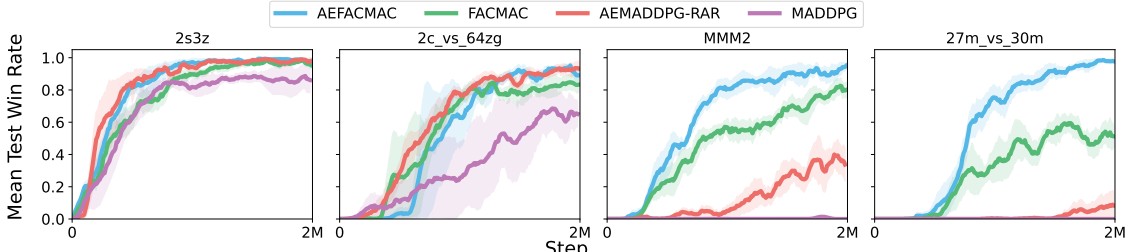

*Figure 5.* Mean test win rate of AEFACMAC, AEMADDPG-RAR, FACMAC and MADDPG on SMAC.

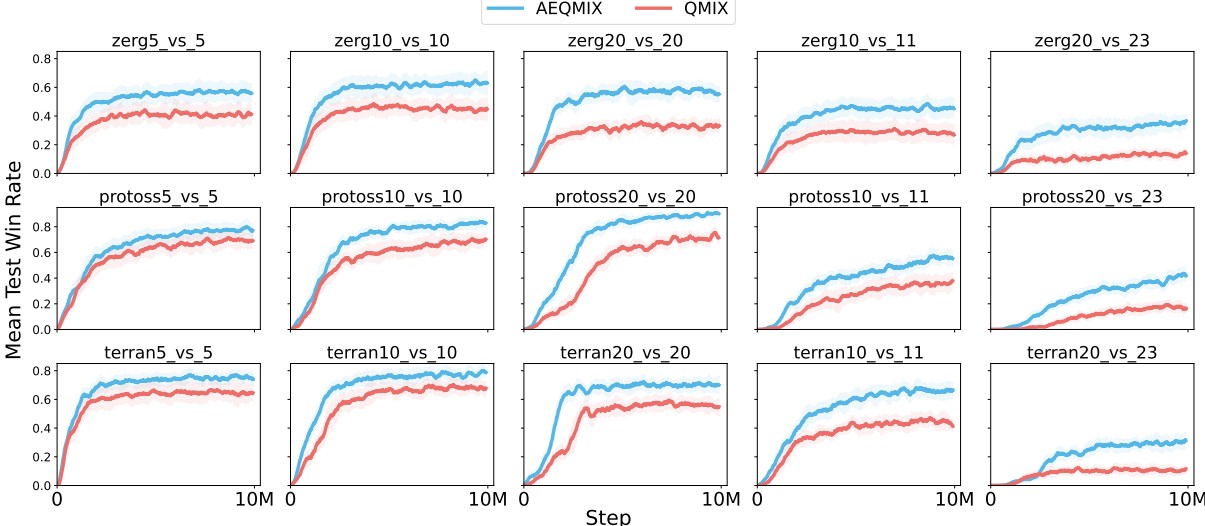

*Figure 6.* Mean test win rate of AEQMIX and QMIX on SMACv2.

## 5. Experiments

In this section, we evaluate the performance of the proposed approach across several popular domains, including SMAC (Samvelyan et al., 2019) and SMACv2 (Ellis et al., 2023). Due to space limitations, we primarily focus on QMIX (Rashid et al., 2020b) in SMACv2, and FACMAC (Peng et al., 2021) and MADDPG (Lowe et al., 2017) in SMAC, as they represent value-based and policy-based off-policy MARL methods, respectively. We refer to the algorithms with annealed multi-step bootstrapping and averaged TD-target as Annealed Ensemble QMIX (AEQMIX), AEFAC-MAC, and AEMADDPG-RAR. Further results, including AEQMIX and QMIX on SMAC and GRF (Kurach et al., 2020), AEVDN, AEQPLEX-RAR, VDN (Sunehag et al., 2018), and QPLEX-RAR on SMACv2, as well as details regarding hyperparameters and experimental settings, can be found in Appendix D.

### 5.1. Main Results

**SMAC**. We evaluate on four maps from the FACMAC paper (Peng et al., 2021), including one Easy map, one Hard map,

and two Super Hard maps. The codebases for FACMAC and MADDPG are adopted from Peng et al. (2021). The results are shown in Fig. 5, where we observe that AEFACMAC and AEMADDPG-RAR significantly improve performance over their base algorithms. The performance gain is particularly notable in the two Super Hard maps with larger joint action spaces, indicating that our methods effectively address the challenges posed by the increased joint action space. Despite MADDPG's generally poor performance in SMAC, our approach allows it to perform well on the Easy and Hard maps and even enables learning on the Super Hard maps, where MADDPG typically struggles.

**SMACv2**. SMACv2 was introduced to overcome some of the limitations of SMAC, such as the lack of stochasticity and partial observability (Ellis et al., 2023). Unlike SMAC, SMACv2 incorporates randomly generated units and initial positions, which introduce more stochasticity and create challenging scenarios. We evaluate QMIX on SMACv2, as it is the SOTA method for this domain but still requires improvement, especially for tasks with large agent numbers. The codebase is from Pymarl2 (Hu et al., 2023). As shown in Fig. 6, we test the algorithms on 15

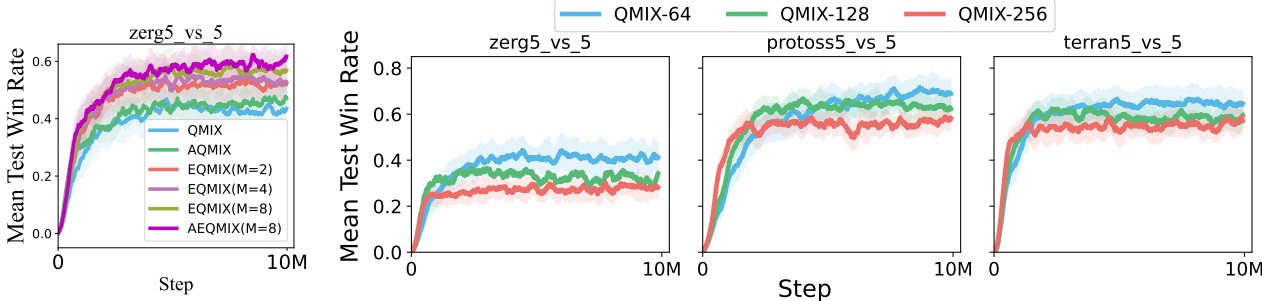

*Figure 7.* Ablations of QMIX with different $\lambda$, $M$ and *hidden_size*.

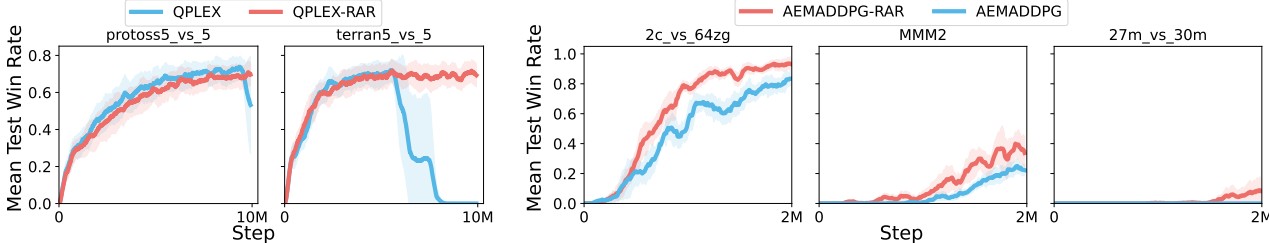

*Figure 8.* The effect of RAR on QPLEX and MADDPG.

maps of SMACv2. QMIX struggles to achieve high win rates across several tasks, particularly in the *10_vs_11* and *20_vs_23* scenarios. However, AEQMIX exhibits significant improvements across all maps, especially on the more challenging tasks where QMIX's win rate is notably low. These results demonstrate the effectiveness of our proposed method in improving existing SOTA algorithms and tackling challenging environments.

## 5.2. Ablation Studies and Discussions

In this subsection, we conduct further experiments to analyze the impact of the techniques introduced in our paper.

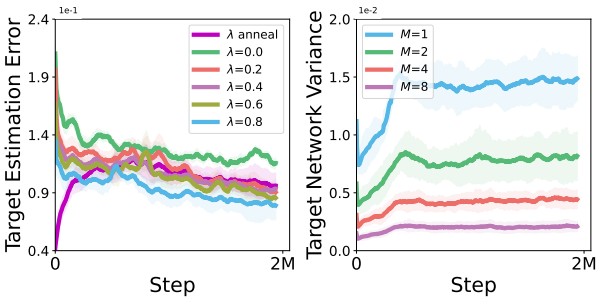

*Figure 9.* TEE and its variance with different $\lambda$ and $M$.

Figure 9 shows the impact of varying $\lambda$ and ensemble size $M$ on TEE. As expected, a larger $\lambda$ results in lower TEE. Our $\lambda$ annealing approach keeps TEE low throughout training, promoting efficient learning in the early stages while mitigating bias in the later stages. Additionally, increasing the ensemble size $M$ reduces the variance, consistent

with the well-established property of ensemble methods to decrease variance by a factor of $1/M$.

Figure 7 (left) illustrates the effects of ensemble size and annealing on performance, where AQMIX represents QMIX with annealing, and EQMIX represents AEQMIX without annealing but with ensemble size $M$. First, we observe that larger ensemble sizes consistently improve performance, demonstrating the benefits of variance reduction through averaged targets. Second, comparing AEQMIX with EQMIX under the same $M$ shows that annealing generally enhances performance. Finally, the combined approach of annealing and averaging results in a mutually beneficial effect, leading to significant performance improvements. It is worth noting that simply increasing the number of parameters does not guarantee better performance. For example, Figure 7 (right) shows the performance of QMIX with varying *hidden_size* of its networks. Here, performance worsens as the network size increases, suggesting that larger networks are more prone to extrapolation errors. This implies that performance gains arise not from an increase in model parameters, but from more accurate target estimation.

Figure 8 shows the effect of RAR on QPLEX and MADDPG. By restricting the action representation, the performance of QPLEX stabilizes, and MADDPG's performance improves. While this technique limits the expressive power of the action space, it highlights that full expressiveness is not always necessary, especially when the joint action space is large. In such cases, overly expressive models may suffer from estimation errors, negatively impacting performance.

# 6. Conclusion

This paper highlights the often-overlooked issue of TEE in off-policy MARL. We identify that TEE primarily arises from extrapolation errors in the large joint action space of the joint Q-function and provide a detailed analysis of its propagation and reduction. We propose three techniques to mitigate TEE from different perspectives, which are broadly applicable to a wide range of off-policy MARL methods. Our experimental results demonstrate that the proposed methods can significantly improve the performance of both policy-based and value-based off-policy MARL algorithms, including factorized and non-factorized approaches. We believe that our findings open up new avenues for future research and offer a fresh perspective on addressing the challenges of TEE in off-policy MARL.

## Acknowledgements

This work was supported by NSFC under Grant 62476008. The authors would like to thank the anonymous reviewers for their valuable comments and advice.

## Impact Statement

There are many potential societal consequences of our work, none which we feel must be specifically highlighted here.

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

## A. Related Works

**Value factorization**. In addition to the previously discussed methods, various approaches address the challenge of value factorization. For value-based methods, Qatten (Yang et al., 2020) employs an attention mechanism to augment the expressive capacity of linear factorization. WQMIX (Rashid et al., 2020a) improves QMIX's expressive ability through the incorporation of a weighted operator and a true value network. QTRAN++ (Son et al., 2020) refines the constraints of QTRAN to improve efficiency. For policy-based methods, VMIX (Su et al., 2021) applies QMIX's factorization to the value function of A2C (Mnih et al., 2016). DOP (Wang et al., 2021c) utilizes linear factorization on the Q-function of COMA (Foerster et al., 2018), while FOP (Zhang et al., 2021b) extend QPLEX's factorization to soft actor-critic (Haarnoja et al., 2018) framework. FACMAC (Peng et al., 2021) combines QMIX's factorization with MADDPG (Lowe et al., 2017). These approaches primarily concentrate on enhancing factorization itself, specifically addressing the TAE problem introduced in this paper, without delving into the underlying reasons for the success of value factorization. While theoretical papers such as (Wang et al., 2021a) take steps to unveil the efficiency and credit assignment of value factorization, they lack substantial support for subsequent improvements. Recent examinations of these methods (Yu et al., 2022; Ellis et al., 2023; Hu et al., 2023), along with more comprehensive experiments, highlight QMIX as the most popular and robust value-based MARL algorithm. Therefore, distinct from previous approaches, our work approaches value factorization from a novel perspective, introducing further enhancements to existing methods.

**Ensemble RL and MARL**. Our analysis is similar to Averaged-DQN (Anschel et al., 2017), which ensembles the Q-functions from the past $M$ steps. Despite proving effective in variance reduction, Averaged-DQN relies on assumptions that may not always hold in practice. Other ensemble methods (Lee et al., 2021) incorporating the standard deviation of Q-functions were not discussed here due to limited observed improvements and their divergence from the main focus of this paper. REDQ (Chen et al., 2021) employs in-target minimization across a random subset of Q-functions from the ensemble. However, this approach proves unsuitable for value factorization, possibly due to the presence of model bias (See Appendix D.2). In MARL, EMAX (Schäfer et al., 2023) applied a similar ensemble method on VDN and QMIX with UCB and majority vote to improve exploration. MMD-MIX (Xu et al., 2021) introduce REM (Agarwal et al., 2020) into a distributional view of QMIX to be more robust in randomness. These methods do not explicitly consider the extrapolation error.

**Offline RL**. The in-target average ensemble employed in our paper bears resemblance to the approach used in offline RL (Agarwal et al., 2020; Fujimoto et al., 2019; Levine et al., 2020). Additionally, the out-of-distribution (OOD) action studied in offline RL aligns with the extrapolation error addressed in this paper. As a result, similar methods may yield comparable effects. However, different from An et al. (2021); Bai et al. (2022), our paper does not require a more conservative/pessimistic estimation of the target, as the monotonic constraint enables self-correction in online RL. Moreover, we found that any degree of pessimism negatively impacts performance, as detailed in the Appendix D.2.

## B. Proof of Proposition 4.1

**Lemma B.1** ((Harutyunyan et al., 2016)). *The PQL operator can be rewritten in the following forms:*

$$\mathcal{N}_{\lambda}^{\mu,\pi}Q = (\mathcal{I} - \gamma\lambda\mathcal{P}^{\mu})^{-1}(r + \gamma(1-\lambda)\mathcal{P}^{\pi}Q). \tag{10}$$

Using this lemma, we have:

$$\mathcal{N}_{\lambda}^{\mu,\pi}(Q_k + e_k) = (\mathcal{I} - \gamma\lambda\mathcal{P}^{\mu})^{-1}[r + \gamma(1-\lambda)\mathcal{P}^{\pi}(Q_k + e_k)]$$
$$= \mathcal{N}_{\lambda}^{\mu,\pi}Q_k + \gamma(1-\lambda)(\mathcal{I} - \gamma\lambda\mathcal{P}^{\mu})^{-1}\mathcal{P}^{\pi}e_k.$$

As a result,

$$\|Q_{k+1} - \mathcal{N}_{\lambda}^{\mu,\pi}Q_k\|_{\infty} = \gamma(1-\lambda)\|(\mathcal{I} - \gamma\lambda\mathcal{P}^{\mu})^{-1}\mathcal{P}^{\pi}e_k\|_{\infty}$$
$$\leq \frac{\gamma(1-\lambda)}{1-\gamma\lambda}\|e_k\|_{\infty} = \beta\epsilon.$$

This shows the propagation of TEE relative to $\lambda$ on each step.

For the algorithm:

$$\pi_k \in \boldsymbol{G}(Q_k) \ and \ Q_{k+1} = \mathcal{N}_{\lambda}^{\mu,\pi_k}Q_k + \varepsilon_k,$$

---

**Algorithm 1** AEQMIX

---

1: Initialize $M$ action-value networks for all agents $\{[Q_i(\tau_i, a_i; \theta_i^j)]_{j=1}^M\}_{i=1}^n$ with parameter $\boldsymbol{\theta}$ and a mixing hypernetwork $H$ with parameter $\psi$
2: Initialize target networks: $\psi' = \psi$, $\boldsymbol{\theta}' = \boldsymbol{\theta}$
3: Initialize replay buffer $\mathcal{D} = \{\}$
4: **while** $k \leq episode\_max$ **do**
5:     set trajectory buffer $T = [\,]$
6:     **for** $t = 1$ to $max\_epsode\_length$ **do**
7:         Explore using $\varepsilon - greedy$ with $\bar{Q}_i(\tau_i, \cdot) = \sum_{j=1}^M Q_i(\tau_i, \cdot; \theta_i^j)$
8:         Store transition $(s_t, \boldsymbol{\tau}_t, \boldsymbol{a}_t, r_t, s_{t+1}, \boldsymbol{\tau}_{t+1})$ into $T$
9:     **end for**
10:     Store trajectory into $\mathcal{D}$ and sample a mini-batch $b$
11:     **for** each trajectory $T$ in $b$ **do**
12:         **for** each transition $(s, \boldsymbol{\tau}, \boldsymbol{a}, r, s', \boldsymbol{\tau}')$ in $T$ **do**
13:             Form joint action $\boldsymbol{a}'$ by $a_i' = \arg\max \bar{Q}_i(\tau_i, \cdot)$
14:             Compute target joint value $\bar{Q}'(s', \boldsymbol{a}')$ using **(??)**
15:         **end for**
16:         Compute PQL target $y_{s,a}$ with $\lambda$ using $\bar{Q}'(s', \boldsymbol{a}')$
17:         Compute joint value $Q(s, \boldsymbol{a}; \boldsymbol{\theta}^j, \psi)$
18:     **end for**
19:     Compute loss through (9)
20:     Adam updates $\boldsymbol{\theta}$, $\psi$ with the computed loss
21:     **if** $k \bmod d = 0$ **then**
22:         Update target networks: $\psi' = \psi$, $\boldsymbol{\theta}' = \boldsymbol{\theta}$
23:         Update $\lambda$ through (13)
24:     **end if**
25:     $k = k + 1$
26: **end while**

---

(Kozuno et al., 2021) introduced the following lemma:

**Lemma B.2** ((Kozuno et al., 2021)). *For any $K$ the following holds:*

$$\|V^{\rho\dagger} - V^{\rho_K}\| \leq \mathcal{O}(\beta^K) + \frac{2}{1-\gamma} \sum_{k=0}^{K-1} \beta^{K-k-1} \|\varepsilon_k\|_\infty \tag{11}$$

*where $\rho_K = \lambda\mu + (1-\lambda)\pi_k$, $\rho_\dagger = \lambda\mu + (1-\lambda)\pi_\dagger$ and $\pi_\dagger \in \boldsymbol{G}(Q^{\rho\dagger})$.*

Therefore, in this paper, we have

$$\|V^{\rho\dagger} - V^{\rho_K}\| \leq \mathcal{O}(\beta^K) + \frac{2}{1-\gamma} \sum_{k=0}^{K-1} \beta^{K-k-1} \cdot \beta\epsilon \tag{12}$$

The second term represents the error dependence which can be futher written as:

$$\frac{2}{1-\gamma} \sum_{k=0}^{K-1} \beta^{K-k} \epsilon = \frac{2\beta}{1-\gamma} \frac{1-\beta^K}{1-\beta} \epsilon = \mathcal{O}\left(\frac{\gamma(1-\lambda)}{(1-\gamma)^2} \epsilon\right).$$

This completes the proof.

## C. Pseudo code

The pseudo code of AEQMIX is summarized in Algorithm 1.

# D. Experimental Details

## D.1. Hyperparameters

Our implementation of VDN, QMIX and QPLEX is based on the pymarl2 (Hu et al., 2023) code base. All hyper-parameters used in our algorithm is consistent with QMIX except for the additional $\lambda^*$ and ensemble size $M$, as presented in Table 1. For stability, $\lambda_k$ is updated alongside the fixed target network as follows:

$$\lambda_k = \lambda^* + \frac{1 - \lambda^*}{1 + \alpha k}, \tag{13}$$

where $\alpha = 10/T$, and $T$ is the total number of environmental steps for training. While this scheme is heuristic, the algorithm is not sensitive to the specific annealing schedule, as long as $\lambda$ converges to $\lambda^*$. The implementation of MADDPG and FACMAC is directly taken from Peng et al. (2021), without any parameter adjustment.

Table 1. Hyperparameters used for SMAC, SMACv2 and GRF.

| hyperparameters | SMAC | SMACv2 | GRF |
| --- | --- | --- | --- |
| Action Selector | epsilon greedy | epsilon greedy | epsilon greedy |
| $\epsilon$ start | 1.0 | 1.0 | 1.0 |
| $\epsilon$ finish | 0.05 | 0.05 | 0.05 |
| $\epsilon$ Anneal Time | 100000 | 100000 | 100000 |
| Runner | parallel | parallel | parallel |
| Batch Size Run | 8 | 4 | 32 |
| Buffer Size | 5000 | 5000 | 2000 |
| Batch Size | 128 | 128 | 128 |
| Optimizer | Adam | Adam | Adam |
| Target Update Interval | 200 | 200 | 200 |
| Mixing Embed Dimension | 32 | 32 | 32 |
| Hypernet Embed Dimension | 64 | 64 | 64 |
| Learning Rate | 0.001 | 0.001 | 0.0005 |
| $\lambda$ | 0.6 | 0.4 | 0.8 |
| $\lambda^*$ | {0.0, 0.4} | {0.0, 0.2} | 0.8 |
| Ensemble Size | 8 | 8 | 2 |
| Gamma | 0.99 | 0.99 | 0.999 |
| RNN Hidden Dim | 64 | 64 | 256 |

## D.2. Additional Discussion on Averaged Target

In this section, we provide further discussion on the averaged Q-target.

First, we examine the effect of using multiple mixing networks to compute the ensembled target:

$$Q(s, \boldsymbol{a}; \boldsymbol{\theta}, \bar{\psi}) = \sum_{j=1}^{M} H_j(s, Q_1(s, a_1; \theta_1), ..., Q_n(s, a_n; \theta_n); \psi^j). \tag{14}$$

As shown in Fig. 10(a), using more than one mixing network ($M > 1$) does not improve performance. This is because the error predominantly arises from the action space, while the mixing network only takes the state as input, which does not effectively address the action-related errors.

Next, we explore whether using a more conservative TD target could improve performance, similar to the approaches used to address extrapolation error in offline RL (An et al., 2021; Bai et al., 2022). To this end, we implement a target similar to REDQ (Chen et al., 2021):

$$y = r + \gamma \min_{j \in \mathcal{M}} Q(s, \boldsymbol{a}; \boldsymbol{\theta}^j, \psi) \tag{15}$$

where $\mathcal{M}$ is a set of $M$ distinct indices from the ensemble $1, 2, \ldots, 10$. The results, shown in Fig. 10(b), represent the win rates averaged across *zerg_5_vs_5*, *protoss_5_vs_5* and *terran_5_vs_5* on SMACv2 at 3M time steps. As the level of

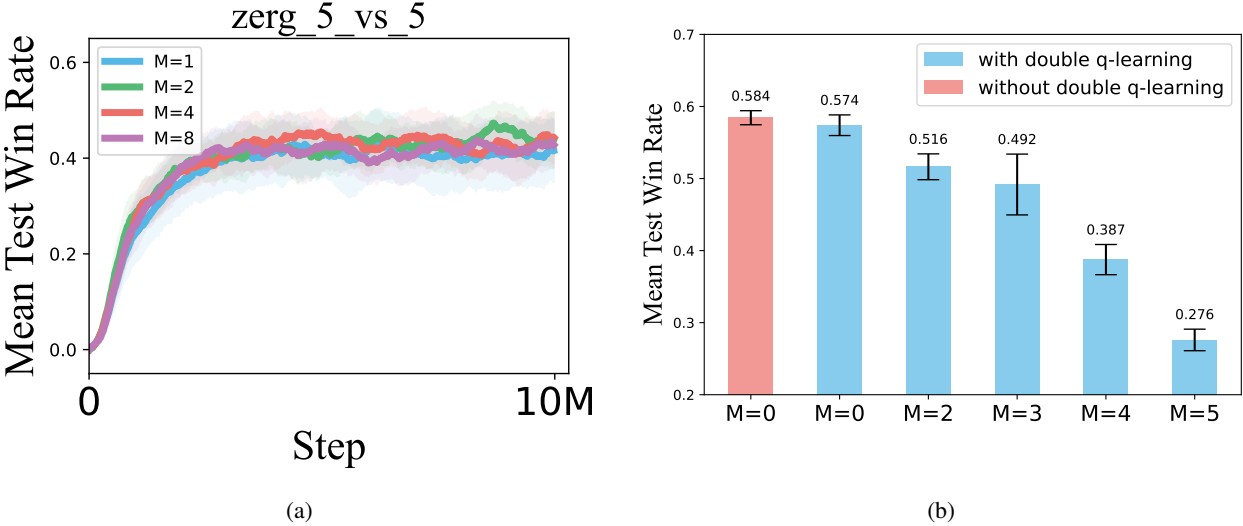

*Figure 10.* (a) Performance of QMIX with different number of mixing network. (b) Comparison of the performance with different pessimism of the TD target.

pessimism increases along the x-axis, we observe that using a more pessimistic TD target does not improve performance and, in fact, has a negative impact. This is in contrast to offline RL settings, where pessimism has proven useful for mitigating extrapolation errors. However, in our online RL setting, this approach introduces unnecessary conservatism, which hinders learning and leads to suboptimal performance.

### D.3. Additional Results on SMAC, SMACv2 and GRF

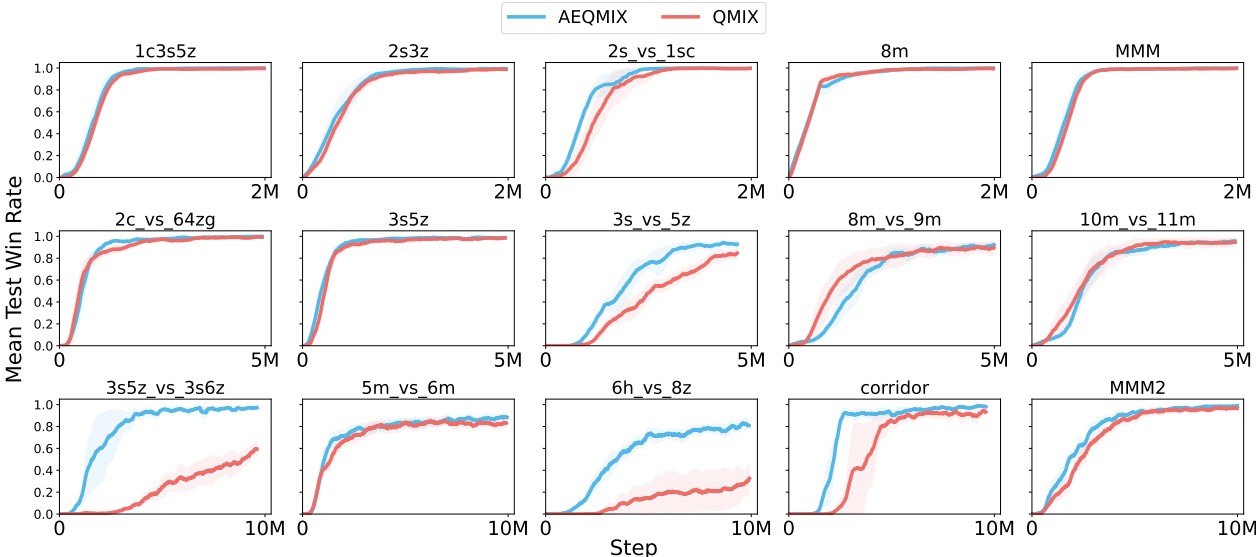

*Figure 11.* Mean Test Win Rate of AEQMIX and QMIX on SMAC.

Fig. 11 shows the performance of AEQMIX and QMIX on SMAC. Fig. 12 shows the performance of AEQPLEX-RAR, QPLEX-RAR, AEVDN and VDN on SMACv2. Fig. 13 shows the performance of AEQMIX and QMIX on GRF.

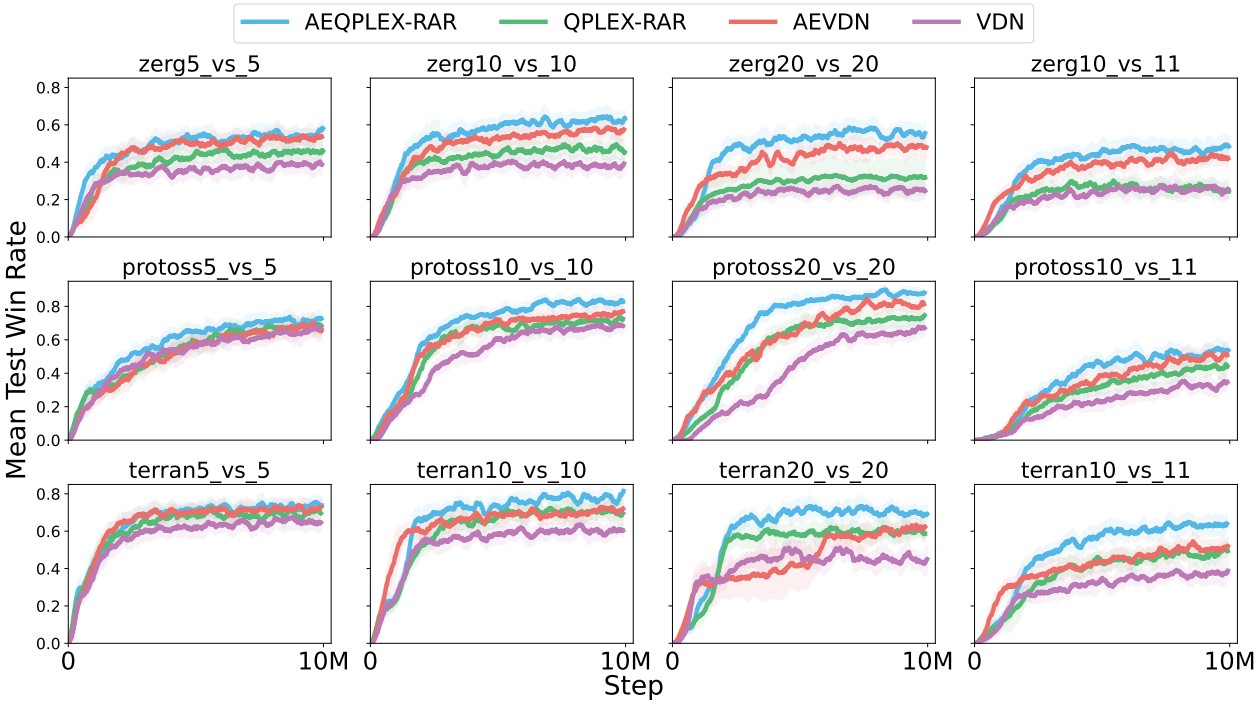

*Figure 12.* Mean Test Win Rate of AEQPLEX-RAR, QPLEX-RAR, AEVDN and VDN on SMACv2.

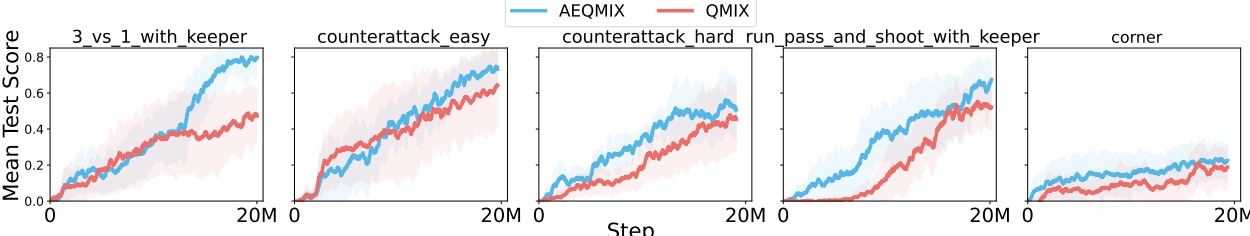

*Figure 13.* Mean Test Score of AEQMIX and QMIX on GRF.

