# OpenReview forum: "Revisiting Cooperative Off-Policy Multi-Agent Reinforcement Learning"
_ICML.cc/2025/Conference — ICML 2025 poster_

### Official Review · Reviewer_Lbue · 2025-03-11

**Overall Recommendation:** 4

**Summary:**

This paper studies the extrapolation error in off-policy multi-agent reinforcement learning that is caused by the curse of multi-agent. To mitigate the errors, the paper proposed to focus on target estimation error (TEE). To further address the TEE, 3 different approaches, annealed multi-step bootstrapping, average TD target and restricted action representation (RAR), are proposed. Empirical simulations show that the proposed approaches show promising results in comparison with their vanilla counterparts.

**Claims And Evidence:**

Yes.

**Essential References Not Discussed:**

The reviewer believes related works are sufficient.

**Experimental Designs Or Analyses:**

The reviewer didn’t implement the pseudo code in the local machine to redo the experimental studies, but went over the simulation results.

**Methods And Evaluation Criteria:**

Proposed methods make sense.

**Other Comments Or Suggestions:**

1.	Some detailed explanations or analysis are needed, for example on why RAR versions are outperforming their counterparts in Figure 4-6.
2.	Among the three approaches, it will be more beneficial to understand and compare the contributions of each induvial approach.
3.	It seems like some of the aspects studied in the paper even applies to on-policy setting in MARL, will extrapolation error also be effectively mitigated by the proposed approaches?

**Other Strengths And Weaknesses:**

Strength: The paper studied the large action space issues in multi-agent system. By focusing on TEE, the paper proposed 3 different approaches, among which RAR is the most interesting to the reviewer. The empirical results show promising results for these approaches in comparison with their vanilla counterparts.
Weakness: Please see the comments sections.

**Questions For Authors:**

Please see the comment section.

**Relation To Broader Scientific Literature:**

The paper proposed 3 different approaches in addressing the large action space issue in multi-agent setting. These approaches, notably including restricted action representation, might provide inspirations in large action space problems.

**Theoretical Claims:**

The reviewer didn’t check the correctness of the proofs.

---

> ### Author Rebuttal · Authors · 2025-03-31
>
> We are thankful for your time and effort in reviewing our paper, which has greatly helped us improve the quality of our paper. We were glad to hear that you found our proposed methods sensible and that our empirical results demonstrated promising improvements over baseline approaches.
> Below, we address the key concerns you raised.
>
> > 1. Some detailed explanations or analysis are needed, for example on why RAR versions are outperforming their counterparts in Figure 4-6.
>
> The superior performance of RAR versions can be attributed to the reduction in Target Estimation Error (TEE) when the joint action space is mapped into a lower-dimensional space. To validate this, we conducted experiments with MADDPG-RAR using different joint action dimensions and report the corresponding TEE values in the following table:
>
> | Action Dim \ Step | 0M |1M|2M|3M|4M|5M|
> | - | - | - | - | - | - | - |
> | 4  | 0.13 | 0.09| 0.09|0.07|0.05|0.04|
> | 16 | 0.17| 0.11| 0.10| 0.08|0.06|0.06|
> | 64 | 0.26| 0.10| 0.09| 0.10|0.08|0.09|
> | 256| 0.27| 0.15| 0.11| 0.09|0.09|0.08|
> |original MADDPG(3125)|0.36|0.18|0.13|0.13|0.11|0.10|
>
> From these results, we observe that a lower action dimension leads to a smaller TEE. This occurs because reducing the joint action space mitigates extrapolation in the target Q-function.
>  Intuitively, if the joint action is mapped into a single dimension (although suboptimal due to potential bias and increased optimality difference, as discussed in Section 3.1), the target Q-function would require no extrapolation even for unseen actions. By properly constraining the joint action dimension, we can significantly reduce extrapolation while maintaining sufficient expressiveness in the joint Q-function, ultimately improving performance.
>
> Another key intuition is that joint Q-functions do not necessarily need to assign distinct values to every possible joint action. For example, consider two agents A and B:
>
> -	Joint action $a_1$: A moves toward B, B stays.
> -	Joint action $a_2$: A stays, B moves toward A.
>
> If the task depends only on the relative distance between A and B, then $a_1$ and $a_2$ may be equivalent in terms of their Q-value. Now, assume:
>
> -	$a_1$ and $a_2$ is frequently observed in state $s_1$.
> -	$a_1$ is frequently observed in state $s_2$, but $a_2$ is not.
>
> In an unrestricted joint Q-function, $Q(s_2,a_2)$ may be poorly estimated due to lack of experience. In contrast, with RAR, since the model has already learned in $s_1$ that $a_1$ and $a_2$ are equivalent, and since $Q(s_2,a_1)$ is well estimated, $Q(s_2,a_2)$ will also be well estimated. This further supports why RAR enhances performance.
>
> > 2. Among the three approaches, it will be more beneficial to understand and compare the contributions of each induvial approach.
>
> Figure 7 (left) illustrates the contributions of Annealed Multi-Step Bootstrapping and Averaged TD Target to QMIX. To further clarify their impact, we provide additional results:
>
> | algorithm\map     | zerg_5_vs_5| zerg_10_vs_10| zerg_10_vs_11| zerg_20_vs_20| protoss_5_vs_5| terran_5_vs_5| terran_10_vs_10| terran_10_vs_11|
> | - | - | - | - | - | - | - | - |-|
> |AEQMIX|62.1|64.4|47.4|56.0|78.1|76.2|80.3|68.9|
> |EQMIX|57.8|62.6|43.4|55.1|75.3|75.1|78.3|63.8|
> |QMIX|40.4|45.0|26.5|33.1|69.5|64.4|66.6|40.7|
>
> These results indicate that both Annealed Multi-Step Bootstrapping and Averaged TD Target contribute to performance improvement. However, Averaged TD Target provides a greater benefit at the cost of additional network computations, whereas Annealed Multi-Step Bootstrapping is more computationally efficient.
>
> The impact of RAR techniques on performance is shown in Figures 4 and 8, where we compare MADDPG and QPLEX with and without RAR. Overall, these results reinforce that all three approaches contribute to performance gains.
>
>
>
> > 3. It seems like some of the aspects studied in the paper even applies to on-policy setting in MARL, will extrapolation error also be effectively mitigated by the proposed approaches?
>
> While some insights from our work could extend to on-policy MARL, extrapolation error is less of a concern in such settings because on-policy methods primarily use a value function (V-function) rather than a Q-function. Specifically:
>
> - Annealed Multi-Step Bootstrapping: The on-policy equivalent is TD($\lambda$) which is already widely used. Unlike in off-policy RL, annealing $\lambda$ in TD($\lambda$) is unnecessary because it does not introduce bias as in Q($\lambda$).
>
> - Averaged TD Target: While applicable to on-policy RL, its effectiveness is reduced. The V-function is inherently easier to learn than the Q-function and is less affected by variance induced by the joint action space.
>
> - Restricted Action Representation: This technique is not applicable to on-policy RL, as it specifically targets the joint action space, which is not used in V-function-based methods.

---

> > ### Comment · Reviewer_Lbue · 2025-04-08
> >
> > I would like to thank the authors for the clarification and good response. I have increased the score accordingly.

---

### Official Review · Reviewer_o7jb · 2025-03-12

**Overall Recommendation:** 3

**Summary:**

The work studies the problem of overestimation and target estimation errors in off-policy multi-agent reinforcement learning (MARL). The work first outlines how action-value estimation in MARL often suffers from estimation errors as a consequence of the exponential growth of the joint action space. To substantiate the problem, the work decomposes the value error into three components and discusses how in particular target estimation error (TEE) can cause worse performance in MARL, and how error propagates in value decomposition algorithms. The work proposes three approaches to mitigate such target estimation error based on (1) multi-step target estimation, (2) computing averaged target values across an ensemble of value functions, and (3) projecting the large action space into lower dimensional discrete space to simplify learning action-value functions. The efficacy of these components applied to multiple MARL algorithms in FACMAC, MADDPG, QMIX, and QPLEX are demonstrated in empirical evaluations across several tasks of the SMAC, SMACv2 and GRF benchmarks. Lastly, the work shows further analysis and ablations to show the impact of each of the three novel proposed components.

**Claims And Evidence:**

Overall, the claims in this work are mostly well supported and convincing. However, the key claim that all three proposed techniques mitigate TEE (e.g. made in Conclusion) is insufficiently supported. The evaluation clearly demonstrates that all three proposed techniques improve the performance of off-policy MARL algorithms, but only for annealed multi-step boostrapping do the authors provide explicit analysis and show how this reduces TEE (Figure 9 left). Given the claims made, I would expect clear empirical evidence that averaged TD targets and restricted action representations reduce TEE.

**Essential References Not Discussed:**

I am not aware of any essential references that are not discussed.

**Experimental Designs Or Analyses:**

1. Figure 9 (left) clearly shows the impact of the proposed $\lambda$ annealing and choice in general on the TEE. However, similar visualisation is missing for the averaged TD target computation and restricted action representation. To support the claims made in this work, I would expect to see similar visualisations to Figure 9 (left) for e.g. AQMIX/ AMADDPG with different $M$, and for MADDPG/ FACMAC/ QPLEX with and without RAR.
2. Figure 7 (left) compares the win rates of QMIX, AEQMIX and different ablations of the AEQMIX algorithm. While EQMIX with varying $M$ all seem to significantly improve upon QMIX, no significant performance difference can be observed between EQMIX ($M=8$) and AEQMIX ($M=8$) as claimed in the text. Would the authors be able to show such ablation potentially for another task and/ or aggregated across multiple tasks to show that AEQMIX indeed performs better than EQMIX?
3. In Section 4.3 and Section 5, empirical evidence is provided that the restricted action representation is improving the performance of the MADDPG and QPLEX algorithms. However, there is a lack of analysis to show how the RAR technique affects the target estimation error and what is being learning. In particular, I would suggest the following analysis:
	1. MADDPG-RAR: Would the authors be able to visualise or otherwise provide intuition into the learning low-dimensional representation of the high-dimensional action space represented by function $g$? Which joint actions are mapped to the same representation and which joint actions are separated?
	2. QPLEX-RAR: Figure 4 visualises the $\lambda$ values for QPLEX. Would the authors be able to show similar values for QPLEX-RAR when applying the Sigmoid function? I know that these values would be within [0, 1] but it is not clear to me whether these would remain more stable or change similarly erratic as for QPLEX.
	3. Impact on target estimation error: The work claims that all three proposed techniques (including RAR) mitigate TEE, so I would expect explicit analysis showing how MADDPG-RAR and QPLEX-RAR exhibit lower TEE than their vanilla counterparts.
4. Figure 1 nicely illustrates the challenge of learning action-value functions in MARL in tasks with many agents. How would AE- and RAR-extended algorithms perform in this illustration? Would the take-away still be that algorithms like MAPPO that do not rely on action-value functions are preferred for tasks with many agents or do the proposed approaches bridge that gap?
5. When describing Figure 2, it is stated that the proportion of extrapolated values "is calculated based on the fraction of $(s, a')$ pairs in each update that are absent from the replay buffer". However, just because a state-action pair is absent from the replay buffer does not necessarily mean that it is extrapolation since it could have been trained on earlier during training (unless the replay buffer is large enough to fit all training samples). Because of this discrepancy, I would expect a lower proportion of values to actually be extrapolated than shown in Figure 2 (a). Would the authors be able to compute the true proportions and update the respective Figure?

**Methods And Evaluation Criteria:**

The proposed techniques are sensible and clearly motivated with the problem of target estimation error and value overestimation. The evaluation is well conducted, systematic, and sufficiently extensive.

**Other Comments Or Suggestions:**

I have no further suggestions or comments.

**Other Strengths And Weaknesses:**

I would like to commend the author to a well structured and written paper. I enjoyed reading the work and believe that it naturally presents the problem, supports it with clear visualisations (e.g. Figure 1 and 2) to illustrate and provide evidence, provides supporting theory, and proposes conceptually simple resolutions that are grounded in existing literature.

**Questions For Authors:**

1. Given the claims made by this work, I would expect explicit evidence that shows how each of the proposed components (Annealed multi-step bootstrapping, averaged TD targets, restricted action representations) reduce the TEE but such evidence is only provided for annealed multi-step bootstrapping (Figure 9 left). I would strongly encourage the authors to provide such evidence for the other two techniques, and I will increase my score if this is done convincingly.
2. Would the authors be able to visualise or otherwise provide intuition into the low-dimensional representation learned in MADDPG-RAR of the high-dimensional action space represented by function $g$? Which joint actions are mapped to the same representation and which joint actions are separated?
3. Would the authors be able to show $\lambda$ values for QPLEX-RAR when applying the sigmoid function as proposed, similar as shown in Figure 4 for QPLEX?
4. Figure 1 nicely illustrates the challenge of learning action-value functions in MARL in tasks with many agents. How would AE- and RAR-extended algorithms perform in this illustration?

**Relation To Broader Scientific Literature:**

As stated above, I would have liked to see the discussion of related work in the main body of the paper and not as part of the supplementary material. I would encourage the authors to identify ways of fitting at least some of this within the main work.

**Theoretical Claims:**

I went through the derivations and steps presented in the main paper and found these logical. I did not verify the proofs provided in Appendix B of the supplementary material.

---

> ### Author Rebuttal · Authors · 2025-03-31
>
> Thank you for your constructive feedback, especially for your detailed feedback on the evaluation of TEE and the impact of our proposed techniques. We were glad to hear that you found our work well-structured, clearly motivated, and systematic empirical analysis. Below, we address the key concerns you raised.
>
> > 1. I would expect clear empirical evidence that averaged TD targets and restricted action representations reduce TEE.
>
> For a good estimation of the TD target, both bias and variance play a crucial role. The estimation error of the TD target typically consists of both components. While annealed multi-step bootstrapping reduces bias introduced by extrapolation, it increases variance by assigning higher weight to trajectory returns. Therefore, averaged TD target is importantly utilized to mitigate variance of the target Q-function, and indeed reduced it as shown in Figure 9 (left).
>
> Regarding RAR, it tackles both bias and variance of TEE by directly simplifying the joint Q-function. Intuitively, in an extreme case where the discrete action space is compressed into a single dimension, learning the joint Q-function becomes as simple as learning a V-function (though this introduces a significant optimality gap, as discussed in Section 3.1).
> To further demonstrate the impact of RAR on TEE, we conducted experiments on MADDPG-RAR in MPE. Please refer to point 1 of our rebuttal to reviewer Lbue for details.
>
> > 2. Would the authors be able to show that AEQMIX indeed performs better than EQMIX?
>
> Yes, AEQMIX consistently outperforms EQMIX. Please see point 2 of our rebuttal to reviewer Lbue for supporting results.
>
> > 3. In MADDPG-RAR, Which joint actions are mapped to the same representation and which joint actions are separated?
>
> Strictly speaking, all actions have different representation. This is because $g(a)$ is a multi-categorical distribution, and every joint action can have unique probability.
>
> For example, consider the spread task with 5 agents and each has 5 actions, which maps from a 5^5 joint action space to 2^5. If we directly apply argmax on the probability, the joint actions are divided into 2^5 groups, each group contains between 64 and 127 mapped joint actions. Within each group, however, we did not find significant correlation among these joint actions.
>
> > 4. The $\lambda$ values of QPLEX-RAR when applying the Sigmoid function.
>
> The following table presents $\lambda$ values when applying the Sigmoid function. It shows that $\lambda$ values remain stable throughout training.
>
> | $\lambda$\Step|1M|2M|3M|4M|5M|6M|7M|8M|9M|10M|
> |-|-|-|-|-|-|-|-|-|-|-|
> | $\lambda mean$|0.52|0.59|0.60|0.61|0.62|0.62|0.63|0.64|0.63|0.64|
> | $\lambda max$|0.98|1.00|1.00|1.00|1.00|1.00| 1.00|1.00|1.00|1.00|
> | $\lambda std$|0.16|0.15|0.16|0.17|0.17|0.17|0.17|0.18|0.17|0.18|
>
> > 5. How would AE- and RAR-extended algorithms perform in this illustration? Would the take-away still be that algorithms like MAPPO that do not rely on action-value functions are preferred for tasks with many agents or do the proposed approaches bridge that gap?
>
> Our proposed techniques improve performance, but the gains are not sufficient to match MAPPO. For instance, in the 5-agent scenario shown in Figure 1, MADDPG’s normalized return improves from -60 to -51. However, MAPPO achieves -41, maintaining a performance gap.
>
> This suggests that while our methods enhance Q-function learning, they do not achieve the same scalability as the V-function-based on-policy methods. But this does not mean MAPPO is better in all case, off-policy methods have their own advantages, such as higher sample efficiency.
>
> > 6. Would the authors be able to compute the true proportions of extrapolated values and update the respective Figure?
>
> We apologize for the mistake. The proportion of extrapolated values was not computed from the replay buffer. Instead, we recorded visited state-action pairs using a separate Python dictionary throughout training. As a result, Figure 2(a) reflects the true proportions.
>
> > 7. The work should attempt to make space to discuss related work and literature within the main work.
>
> Thanks for the suggestion. While we included some discussion of related work in Section 2, we acknowledge that it may not be sufficient. We will consider adding a new subsection in Section 2 to better integrate the literature discussion into the main paper.
>
> > 8. The legend of Figure 10 (b) within the supplementary material is misleading.
>
> The term "double Q-learning" in Figure 10(b) refers precisely to [1, 2]. In this figure, the x-axis represents different ensemble sizes, where a larger M leads to a more conservative (underestimated) target. All blue columns apply double Q-learning, which can also contribute to underestimation. To provide contrast, we include a case without double Q-learning when M=0 (pink), which exhibits the most overestimation. The figure demonstrates that simply reducing overestimation does not necessarily lead to better performance.

---

> > ### Comment · Reviewer_o7jb · 2025-04-07
> >
> > I thank the authors for their responses and clarifications.
> >
> > > Therefore, averaged TD target is importantly utilized to mitigate variance of the target Q-function, and indeed reduced it as shown in Figure 9 (left).
> >
> > Do I understand correctly that Figure 9 (left) then shows the TEE for an AE algorithm using both averaged TD targets and multi-step bootstrapped targets using the respective hyperparams? This was not clear to me previously. Also, for which algorithm does Figure 9 show these metrics? Is this for QMIX/ MADDPG/ QPLEX/ ...?
> >
> > > Please refer to point 1 of our rebuttal to reviewer Lbue for details.
> >
> > This provided analysis into the RAR approach is excellent and would expect to see some of it in the revised paper since it provides essential evidence for claims made in the work. The same goes for the provided results for QPLEX-RAR. As a bonus, it would be helpful to also provide TEE values for QPLEX with and without TEE (as done for MADDPG) to showcase that RAR also reduces TEE as claimed since the QPLEX RAR approach differs from the one used in MADDPG.
> >
> > > Our proposed techniques improve performance, but the gains are not sufficient to match MAPPO. For instance, in the 5-agent scenario shown in Figure 1, MADDPG’s normalized return improves from -60 to -51. However, MAPPO achieves -41, maintaining a performance gap.
> > >
> > > This suggests that while our methods enhance Q-function learning, they do not achieve the same scalability as the V-function-based on-policy methods. But this does not mean MAPPO is better in all case, off-policy methods have their own advantages, such as higher sample efficiency.
> >
> > I would really appreciate a mention of this result in the work! In some sense, it can be seen as a negative result but it sheds light on when different algorithms shine which is valuable insights for the MARL community.
> >
> > > The term "double Q-learning" in Figure 10(b) refers precisely to [1, 2]. In this figure, the x-axis represents different ensemble sizes, where a larger M leads to a more conservative (underestimated) target. All blue columns apply double Q-learning, which can also contribute to underestimation.
> >
> > This was not clear to me from the textual description -- a short clarification as provided in the rebuttal would be helpful here.
> >
> > Overall, I remain at my original score and continue to suggest to accept this work -- it provides novel techniques and provides insights that are valuable for off-policy algorithms in MARL.

---

### Official Review · Reviewer_YK2e · 2025-03-13

**Overall Recommendation:** 3

**Summary:**

This paper identifies a problem of erroneous Q-target estimation, primarily caused by extrapolation errors, which worsens as the number of agents increases. The authors follow the previous work on single-agent error decomposition and apply it to multi-agent Q-learning, decomposing the error into Target Approximation Error (TAE), Target Estimation Error (TEE), and Optimality Difference (OD). To address the issue of TEE, the authors propose a suite of techniques, including annealed multi-step bootstrapping, averaged Q-targets, and restricted action representation. Experiments on SMAC, SMACv2, and Google Research Football show significant improvements over baseline methods.

**Claims And Evidence:**

The claims are generally convincing.

**Essential References Not Discussed:**

[1] Regularized Softmax Deep Multi-Agent Q-Learning.

**Experimental Designs Or Analyses:**

- The authors frequently mention the MPE environment in the introduction but do not include any experiments on it. Why was this environment omitted?
- The paper should compare its approach with other research addressing overestimation in MARL, such as [1].

---

[1] Regularized Softmax Deep Multi-Agent Q-Learning.

**Methods And Evaluation Criteria:**

- The proposed methods make sense for addressing the problem of erroneous Q-target estimation.
- The benchmarks, including SMAC, SMACv2, and Google Research Football, are appropriate for assessing the performance of MARL algorithms.

**Other Comments Or Suggestions:**

See Weaknesses

**Other Strengths And Weaknesses:**

Strengths:
- The paper presents several effective improvements over baseline methods.
- The experimental results demonstrate clear performance gains.


Weaknesses:
- The scope of paper is framed around off-policy learning, but the key techniques seem to target algorithms using Bellman optimality equations.
- The section on Restricted Action Representation in MADDPG is difficult to follow. The authors could provide a more intuitive example, such as one where the number of actions differs from the number of agents, rather than using the 5^5 example, which is not clearly explained.
- The paper references the $\lambda$ parameter in QPLEX but does not provide a clear introduction to QPLEX itself. Some background should be added.
- The discussion on  $\lambda$ in QPLEX does not seem to align with the idea of Restricted Action Representation. Instead, it appears to be merely a constraint on the weight values rather than a true action space compression.
- The authors propose applying an additional sigmoid function to $\lambda_i$, but the original QPLEX paper already applies sigmoid before summation. While this paper applies it after summation, the practical impact of this change is unclear.
- The ablation study on Restricted Action Representation is confusing. The results for MADDPG and QPLEX should be presented together in the same figure for easier comparison. Why are the ablation studies conducted in different environments for different methods? This makes it difficult to interpret the results consistently.

**Questions For Authors:**

See Weaknesses

**Relation To Broader Scientific Literature:**

The authors highlight the increasing error in joint-action estimation as the action space expands, which is a well-known issue in multi-agent reinforcement learning. They analyze this issue in detail and propose solutions, making it a relevant and important contribution.

**Theoretical Claims:**

The paper does not introduce particularly complex new theoretical contributions but rather focuses on experimental observations.

---

> ### Author Rebuttal · Authors · 2025-03-31
>
> Thank you for your constructive feedback, which has greatly helped us improve the quality of our paper. We were glad to hear that you found our claims convincing and that our proposed methods make sense for addressing the problem. Below, we address the key concerns you raised.
>
> > 1. The authors frequently mention the MPE environment in the introduction but do not include any experiments on it. Why was this environment omitted?
>
> The MPE environment is included in **Figure 2 (left) and Figure 4 (left)**. However, we did not highlight it in the experiment section because it contains only one suitable task (Spread), and is significantly less complex than SMAC and SMACv2. We will provide additional results for Spread in the appendix in the next version. For futher details, please refer to point 5 of our rebuttal to reviewer o7jb.
>
> > 2. The paper should compare its approach with other research addressing overestimation in MARL, such as [1].
>
> Thank you for the suggestion. We will add a comparison to RES.
> However, based on the current results, it appears that RES does not outperform the fine-tuned QMIX when evaluated within the PyMARL2 codebase.
> This also aligns with our argument that extrapolation error does not always lead to overestimation. As shown in **Figure 10(a) (Appendix D.2)**, applying a more underestimated target actually degrades performance.
>
> > 3. The scope of paper is framed around off-policy learning, but the key techniques seem to target algorithms using Bellman optimality equations.
>
> Our techniques target methods that utilize Q-functions, which are fundamental to most off-policy methods. Since these methods require off-policy Q-target estimation, they inherently face target estimation error, as discussed in section 3 of our paper.
>
> > 4. The section on Restricted Action Representation in MADDPG is difficult to follow. The authors could provide a more intuitive example, such as one where the number of actions differs from the number of agents, rather than using the 5^5 example, which is not clearly explained.
>
> Thanks for this suggestion, we choose the 5^5 to 2^5 example because it directly corresponds to our experimental setup in Figure 4 (left).
> For another example such as 4 agents with 5 action, we can also compress the action space from 5^4 to 2^4.
> For a more detailed explanation of RAR, please refer to point 1 of our rebuttal to reviewer Lbue.
>
> > 5. The paper references the $\lambda$ parameter in QPLEX but does not provide a clear introduction to QPLEX itself. Some background should be added.
>
> Thank you for pointing this out. We will add a brief introduction to QPLEX to provide the necessary background for the discussion on the $\lambda$ parameter.
>
> > 6. The discussion on $\lambda$ in QPLEX does not seem to align with the idea of Restricted Action Representation. Instead, it appears to be merely a constraint on the weight values rather than a true action space compression.
>
> The concept of RAR is to **limit the effect of joint actions on the joint Q-function**.  In QPLEX, since the joint action component is already separated by $\lambda$, directly constraining $\lambda$ can effectively restrict its influence. Applying the same technique as in MADDPG could achieve similar results but would require additional networks. We will provide corresponding experiments in the appendix.
>
> > 7. The authors propose applying an additional sigmoid function to $\lambda_i$, but the original QPLEX paper already applies sigmoid before summation. While this paper applies it after summation, the practical impact of this change is unclear.
>
> The original QPLEX paper applies a sigmoid before summation, but this does not restrict the impact of joint actions, as other weight terms can still amplify it. The full expressive power of QPLEX requires $\lambda\in[0,\infty]$, meaning that even with a pre-summation sigmoid, QPLEX retains full representation capability. By constraining $\lambda\in[0,1]$ , our approach explicitly limits action representation ability, leading to more stable results.
>
> > 8. The ablation study on Restricted Action Representation is confusing. The results for MADDPG and QPLEX should be presented together in the same figure for easier comparison. Why are the ablation studies conducted in different environments for different methods? This makes it difficult to interpret the results consistently.
>
> We understand the concern. However, our MADDPG and QPLEX use **different codebases** as detailed in  Appendix D.1, making direct performance comparison within the same figure difficult (basically because the difference in **parallelized environments**).
> Moreover, the performance of MADDPG and QPLEX are significantly different, SMACv2 may be too hard for MADDPG and SMAC may be too easy for QPLEX, which makes the effect of RAR less noticeable.
> We will clarify this in the next version.

---

> > ### Comment · Reviewer_YK2e · 2025-04-03
> >
> > Thanks for your clarification. I have updated my score. Good luck!

---

### Decision · Program_Chairs · 2025-05-01

**Decision:**

Accept (poster)

**Comment:**

The paper tackles value target estimation errors in off-policy MARL, showing how the large joint action space leads to significant estimation issues. It decomposes value error into three components, highlighting the impact of target estimation error, particularly in value decomposition methods. To address this, the authors propose: (1) multi-step target estimation, (2) ensemble averaging of target values, and (3) projecting actions into a lower-dimensional discrete space. These techniques are evaluated on FACMAC, MADDPG, QMIX, and QPLEX across the SMAC, SMACv2, and GRF benchmarks, with ablation studies. This is a very nice work and all reviewers unanimously recommended for acceptance.